# Self-organization of collective escape in pigeon flocks

**Marina Papadopoulou**[1]*, **Hanno Hildenbrandt**[1], **Daniel W. E. Sankey**[2], **Steven J. Portugal**[3], **Charlotte K. Hemelrijk**[1]

**1** Groningen Institute for Evolutionary Life Sciences, University of Groningen, Groningen, The Netherlands, **2** Centre for Ecology and Conservation, University of Exeter, Penryn, United Kingdom, **3** Department of Biological Sciences, School of Life and Environmental Sciences, Royal Holloway University of London, Egham, United Kingdom

* m.papadopoulou.rug@gmail.com

## Abstract

Bird flocks under predation demonstrate complex patterns of collective escape. These patterns may emerge by self-organization from local interactions among group-members. Computational models have been shown to be valuable for identifying what behavioral rules may govern such interactions among individuals during collective motion. However, our knowledge of such rules for collective escape is limited by the lack of quantitative data on bird flocks under predation in the field. In the present study, we analyze the first GPS trajectories of pigeons in airborne flocks attacked by a robotic falcon in order to build a species-specific model of collective escape. We use our model to examine a recently identified distance-dependent pattern of collective behavior: the closer the prey is to the predator, the higher the frequency with which flock members turn away from it. We first extract from the empirical data of pigeon flocks the characteristics of their shape and internal structure (bearing angle and distance to nearest neighbors). Combining these with information on their coordination from the literature, we build an agent-based model adjusted to pigeons' collective escape. We show that the pattern of turning away from the predator with increased frequency when the predator is closer arises without prey prioritizing escape when the predator is near. Instead, it emerges through self-organization from a behavioral rule to avoid the predator independently of their distance to it. During this self-organization process, we show how flock members increase their consensus over which direction to escape and turn collectively as the predator gets closer. Our results suggest that coordination among flock members, combined with simple escape rules, reduces the cognitive costs of tracking the predator while flocking. Such escape rules that are independent of the distance to the predator can now be investigated in other species. Our study showcases the important role of computational models in the interpretation of empirical findings of collective behavior.

## Author summary

Bird flocks show fascinating patterns of collective motion, particularly when escaping a predator. Little is known, however, about the underlying mechanisms of these patterns.

**Data Availability Statement:** Our agent-based model (HoPE) is available on a GitHub repository at: https://github.com/marinapapa/HoPE-model. All data and code used to produce the results and analyses presented in this manuscript are also

available at: https://github.com/marinapapa/ SelfOrg-ColEsc-Pigeons.

**Funding:** This work has been financed to C.K.H. by the Netherlands Organisation for Scientific Research (NWO - https://www.nwo.nl), the Open Technology Programme (OTP) Preventing bird strikes: Developing RoboFalcons to deter bird flocks project number 14723. The empirical work was funded by a Royal Society (https:// royalsociety.org) Research Grant (R10952) to S.P. The funders had no role in study design, data collection and analysis, decision to publish, or preparation of the manuscript.

**Competing interests:** The authors have declared that no competing interests exist.

We fill this gap by firstly analyzing GPS data of pigeon flocks under attack by a robotic-predator and secondly studying their collective escape in a computer simulation. Previous research on pigeons has revealed that flock members turn away from the predator more the closer the predator gets. Using computer simulations that are based on pigeon-specific characteristics of motion and coordination among individuals, we study what escape rules at the individual level may underlie this distance-dependent pattern. We show that, even if individuals do not intend to escape more when the predator is closer, their escape frequency still increases the closer they get to the predator. This happens by self-organization from the coordination among individuals and despite their tendency to turn away from the predator being distance-independent. A key aspect of this process is the increasing consensus among flock members over the escape direction when the predator gets closer.

## Introduction

Computational models based on self-organization are a valuable tool to disentangle the processes underlying complex patterns of biological systems. Such models show that many collective patterns that are seen in nature may be emergent, i.e. not represented in the behavioral rules of the group members [1]. We present several examples of such emergent phenomena in collective motion below. In simulated fish schools, the oblong shape of the group (common in real schools) emerges from individuals avoiding their nearest neighbor by slowing down or turning away while coordinating [2, 3]. Milling (collective circular motion) emerges in a minimal model of collective motion when individuals are limited in their field of view and angular velocity [4]. In more complex models, changes between milling and schooling behavior at the group level (phase transitions) spontaneously arise from a single set of behavioral rules at the individual level [5, 6]. Realistic group shapes emerge in flocks and schools from the specifics of individuals' locomotion, i.e. flying versus swimming [3]. Swarm shapes also depend on the preceding shape of the group, a phenomenon called hysteresis [5]. In sum, these models help us to determine what behavioral rules at the individual level are necessary for a collective pattern to arise.

Models of collective motion have also been used to study collective patterns of escape from a predator. Inada and Kawachi (2002) developed a two-dimensional model of fish schools under attack. Despite including only one rule of escape at the individual level, several patterns of collective escape emerged while the group was turning, splitting, or surrounding the predator [7]. In airborne flocks of birds, the only pattern of collective escape that has been studied in detail is the agitation wave: a dark band that moves from one side of the group to the other [8–10]. A wave was initially assumed to relate to increased density among individuals when they flee away (accelerating forwards) from the predator and thus come closer to each other [11]. The individual-level rules of this process were studied in a three-dimensional model of European starlings (*Sturnus vulgaris*) [12], where a few initiators were performing an escape motion (either through forward acceleration or turning) that was subsequently copied by their neighbors [13]. In the simulated flocks, a dark band was visible only during the turning escape maneuver because while banking (a motion necessary for birds to turn during flight) the larger surface of the wings becomes visible to an observer. This lead to the conclusion that this band can reflect a wave of orientation instead of density [13]. This study highlighted the importance of such theoretical experiments in the understanding of which behavior underlie complex collective patterns.

Given differences in collective behavior across species [14–16], a computational model should be adjusted to empirical data in order to study in detail specific patterns of groups in nature [12, 17–19]. Collecting quantitative empirical data, however, can be challenging. For

bird flocks, results of field experiments using stereo photography have been extremely valuable for validating model conclusions and formulating hypotheses [12, 14, 16, 20]. Due to the specifics of 3D-imaging techniques, the positions of flock members can be reconstructed only for a few seconds of flight while the flock is passing through a stationary set of cameras. This poses a limitation for studying their collective escape: capturing a full escape sequence within the narrow frame of the cameras is unlikely. Furthermore, the challenge of controlling and tracking an avian predator in the field during a pursuit limits techniques used to collect full trajectories of airborne flocks, such as GPS devices [15, 21]. Due to these constrains, until recently the collective escape of airborne flocks has been empirically studied only through video footage, focusing on qualitative descriptions of the observed collective patterns [8, 10, 22]. This lack of quantitative data has limited our potential to study the underlying processes of collective escape [13, 23].

GPS data of a flock under predation have recently been collected with a robotic falcon attacking flocks of homing pigeons (*Columba livia*) [24]. Based on the tracks of escaping individuals during a pursuit, Sankey *et al.* (2021) [24] identified a new distance-dependent pattern of collective escape: when the predator is close, pigeons are turning away from its heading with higher likelihood than they align with their flockmates. It is however not clear whether this reflects a distance-dependent behavioral rule of individual pigeons or an emergent property.

The aim of the present paper is to study this empirical finding that pigeons in a flock turn away from the predator more frequently when the predator is closer [24]. We do so in a computational model inspired by empirical data [15, 21, 24]. We first analyze the GPS trajectories of individual pigeons and use them to infer their flock shape and internal structure (bearing angle and distance to nearest neighbor) [24]. We combine these data with information on the specifics of flocking in homing pigeons [15, 21, 24] and build a realistic computational model of pigeons' collective escape [17]. We use our model to investigate whether flock members need an individual rule to escape more at closer distances to the predator for their escape frequency to scale with predator-prey distance. We model a predator-avoidance mechanism that reflects our null hypothesis: pigeons turn to escape without taking into account their distance to the predator. We analyze the effect of predator-avoidance and coordination among flock members during an attack. We confirm that our model reproduces the empirical pattern and conclude that the frequency of escape turns increases with decreasing distance to the predator as an emergent property during collective escape.

## Materials and methods

### Flocks of pigeons under attack

**Empirical data.** We used pre-proccessed trajectories of flocks of homing pigeons (*Columba livia*) collected by Sankey *et al.* (2021) [24]. All flock members were trained to fly back to their home after being released at a site approximately 5 km away. A robotic falcon [24, 25], similar to a peregrine falcon (*Falco peregrinus*) in appearance and locomotion, was remotely controlled to attack the flocks after their release and to chase them until they leave the site. Both prey and predator were mounted with GPS devices sampling with a frequency of 0.2 seconds (see [24] for full details).

### A data-inspired computational model

We developed an agent-based model, named *HoPE* (Homing Pigeons Escape), that simulates airborne flocks of pigeons under attack by a predator. We present our models based on elements of the ODD (Overview, Design concepts, Detail) protocol [26].

**Principles, entities, and process overview.** Our model is based on self-organization [1]. It consists of pigeon- and predator-like agents. Each simulation includes several pigeon-agents (also referred to as 'prey') that form a flock, and one predator-agent that pursues and attacks the flock. We adjusted the coordination rules of alignment, avoidance and attraction among nearby pigeons-agents [5, 27, 28] to known behavior of pigeons [15, 21, 24]. Parameters not found in the literature were determined through calibration with empirical measurements of several flock's characteristics, namely the distributions of individuals' speed, nearest-neighbor distance, and relative position of nearest neighbor (bearing angle and distance). All parameters in our model are presented in Table 1 and their values remain constant throughout a simulation.

**Motion and sensing.** Our agents move in a large two-dimensional open space. They are defined by a position vector ($\vec{r}_i$) in the global space, and a velocity ($\vec{v}_i$) and a heading vector ($\hat{h}_i$) in their own coordinate system. We assume that agents always have their heading in the direction of their velocity (non-slip assumption). Each agent senses the position and heading of other agents in its field of view (controlled by the angle $\theta_{FoV}$). The field of view of pigeon-

**Table 1. The parameters of the HoPE model.** The majority of parameter values are taken from previous empirical work on pigeons flocks [15, 21, 24]. We decide the value of parameters that could not be inferred from the literature by calibration [17] through comparisons with the empirical data of Sankey *et al.* (2021) [24] and visual testing to ensure realistic flock formations.

| Parameter | Name-Description | Value | Unit | Reference |
|---|---|---|---|---|
| **Collective motion** | | | | |
| $u_c$ | Mean cruise speed | 16 | *m/s* | [21] |
| $u_c^{sd}$ | Standard deviation of cruise speed | 2 | *m/s* | [21] |
| $m$ | Pigeons' body mass | 0.45 | kg | [21] |
| $\theta_{FoV}$ | Angle of field of view | 215 | degrees | [15] |
| $\theta_{FoV}^f$ | Angle of frontal part of field of view | 180 | degrees | [15] |
| $n_{topo}$ | Number of interacting neighbors for coordination | 7 | individuals | [24] |
| $d_{min}^s$ | Minimum distance for separation force | 1 | m | [15] |
| $d_{min}^\eta$ | Minimum mean distance to neighbors for acceleration | 2 | m | [15] |
| $d_{max}^\eta$ | Maximum mean distance to neighbors for acceleration | 10 | m | [15] |
| $\kappa$ | Strength of deceleration | 0.5 | - | [15] |
| $w_a$ | Weighting factor of alignment force | 10 | - | - |
| $w_s$ | Weighting factor of separation force | 5 | - | - |
| $w_{ct}$ | Weighting factor of turning cohesion force | 2.5 | - | - |
| $w_{ca}$ | Weighting factor of acceleration-based cohesion force | 5 | - | - |
| $w_f$ | Weighting factor of flight-control force | 0.2 | - | - |
| $\beta_w$ | Range of uniform distribution for the weighting factor of the random-error force | 2 | - | - |
| **Collective escape** | | | | |
| $w_p$ | Weighting factor of pigeons' predator avoidance | 2 | - | - |
| $d_{max}^e$ | Minimum distance for pigeons' predator avoidance | 50 | m | [24] |
| $\zeta_p$ | Predator's pursuit speed relative to the flock's | 1.0 | - | - |
| $\xi_p$ | Predator's attack speed relative to the flock's | 1.5 | - | - |
| $d_h$ | Distance of predator's pursuit | 30–40 | m | [24] |
| $\theta_p$ | Bearing angle of predator's pursuit | 130–225 | degrees | - |
| $T_h$ | Predator's pursuit duration | 30 | s | - |
| $T_a$ | Predator's attack duration | 20 | s | - |
| **Simulation** | | | | |
| $dt$ | Integration time | 0.005 | s | - |
| $\Delta t$ | Update frequency | 0.02 | s | - |
| | Sampling frequency | 0.2 | s | [24] |

 

agents is set to $215^o$ and is divided into a front ($\pm90^o$ around their heading, $\theta^f_{FoV}$) and a side area [15].

From all sensed individuals, each agent only interacts with a fixed number of closest neighbors (referred to as 'topological neighbors', $n_{topo}$) [12, 29]. Sankey *et al.* (2021) [24] estimated that the topological range of interaction may differ between small and large flocks, and between alignment and centroid-attraction. Their method is, however, not well established and was unable to reveal the (true) topological range used in our model (S1 Fig). Because of this, and aiming to reduce the complexity of our model [17], we chose a constant number of 7 topological neighbors for both alignment and attraction across different flock sizes [16, 29].

The motion of pigeon-agents in our model is dictated by the sum ($\vec{\psi}_i$) of some external pseudo-forces, namely a coordination force ($\vec{\psi}^{coord}_i$), an escape force ($\vec{\psi}^p_i$), and an internal flight control system ($\vec{\psi}^f_i$). The components of each of these pseudo-forces are shown in Fig 1. Since birds cannot perfectly collect and respond to information about their surroundings (e.g. average position and direction of neighbors), we included a random error in their motion. Specifically, a noise scalar ($\epsilon_i$) is randomly sampled by a uniform distribution (with a range $\beta_w$) and

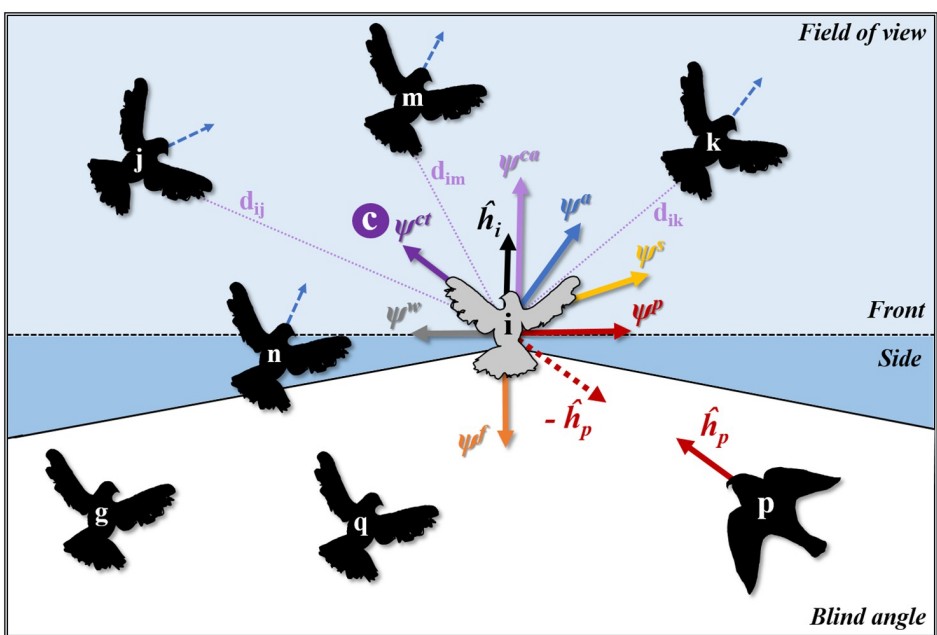

**Fig 1. Collective motion schematic in the computational model *HoPE*.** The colored areas represent the field of view of a focal individual $i$ with heading $\hat{h}_i$, split into a 'front' (light blue, $\theta^f_{FoV}$) and 'side' area (blue). Based on its total field of view ($\theta_{FoV}$), $i$ interacts with its topological neighbors $n$, $j$, $m$, and $k$. Agents $g$ and $q$ are in the blind angle of $i$ and are thus ignored. All pseudo-forces ($\psi$) acting on $i$ are represented by colored arrows. Alignment ($\psi^a$) is the average vector of topological neighbors' headings (indicated by blue doted arrows). Centroid attraction ($\psi^{ct}$) is the vector from $i$'s position to the center of the topological neighborhood ($c$). Accelerating attraction ($\psi^{ca}$) is a vector in the direction of motion (aligned with heading $\hat{h}_i$), depending on the distances of $i$ to the neighbors in its front field of view, namely $j$, $m$ and $k$ (with distances $d_{ij}$, $d_{im}$ and $d_{ik}$ respectively). If there are no agents in the front field of view, then $\psi^{ca}$ is negative. Separation ($\psi^s$) is the vector away from the position of the nearest neighbor $n$. Avoidance of the predator ($\psi^p$) is a vector perpendicular to the heading of the pigeon-agents, in the direction away from the predator's heading ($\hat{h}_p$). In other words, forcing a turn away from the predator's heading clockwise or anticlockwise (left or right) relative to the agent's heading. Vector $\psi^f$ represents the flight-control force that drags $i$ towards its preferred speed. The $\psi^w$ vector is the force to create random-error in the orientation of the agents.

 

multiplied with a unit vector perpendicular to the agents' headings, forming a pseudo-force that affects the agents' turning motion ($\vec{\psi}_i^w = \epsilon_i \hat{h}_i^{\perp}$).

**Individual variation and speed control.** Several models of collective behavior assume constant and often identical speed of all group members [4–6, 19]. Homing pigeons, however, have a preferred 'solo' speed of flight, from which they deviate by up to 2 *m/s* when flying in a flock [21]. When flying in pairs, they accelerate to catch up with their frontal neighbor (the further away the neighbor the higher the acceleration) and slow down if they are in front [15]. According to these findings, agents in our model have different preferred speeds from which they can deviate to stay with their flockmates.

At initialization, the model randomly samples a preferred speed for each agent from a uniform distribution with interval length of 4 *m/s* ($u_c^{sd}$ around the mean $u_c$, Table 1). To reflect the inability of individuals to deviate from their preferred speed for a prolonged period [21], we modeled a drag force ($\vec{\psi}^f$) that pulls agents back to it (Eq 1). This force increases with increasing deviation from the preferred flight speed, according to:

$$\vec{\psi}_i^f(t + dt) = (u_i^{cs} - u_i(t))\ m\ \hat{h}_i(t) \quad \text{Flight-control force} \tag{1}$$

where $u_i^{cs}$ is the personalized speed of agent $i$ with mass $m$, and $u_i(t)$ is its speed on timestep $t$.

**Coordination.** Flock formation emerges from simple rules of among-individuals coordination: attraction, avoidance and alignment [5, 12, 27, 28]. In our model, we parameterized alignment to be the strongest among our steering forces, as found in homing pigeons [24]. The alignment force ($\hat{\psi}^a$) has the direction of the average heading of all topological neighbors. Centroid-attraction in our model is relatively weak, given that coherence in flocks of pigeons is mediated mostly by speed adjustment [15, 24]. We thus introduced an 'acceleration-attraction' force ($\hat{\psi}^{ca}$): individuals accelerate if they have neighbors within their front field of view and decelerate if they do not sense any individuals nearby (their field of view is empty). The strength of this acceleration force increases with increasing average distance to all frontal neighbors, according to a smootherstep function [30] from 2 to 10 m ($d_{min}^{\eta}$ and $d_{max}^{\eta}$) based on [15]. Deceleration is constant to the half of acceleration's maximum (based on the scaling factor $\kappa$) [15]. The centroid-attraction force ($\hat{\psi}^{ct}$) is the unit vector with direction from the position of the focal individual to the average position of its topological neighbors.

Lastly, separation among pigeons is mediated by turning when neighbors are within a 1-meter distance from each other [15]. Similarly in our model, an avoidance force ($\hat{\psi}^s$) pushes agents to turn away from the position of their closest neighbor (instead of all topological neighbors) if they are too close (according to the minimum-separation distance, $d_{min}^s$). We parameterize this switch from 7 to 1 topological range for separation to increase the resemblance of our model to empirical data (following previous theoretical work of [20]), and since there is no, to our knowledge, previous research on the real topological range for separation in pigeon flocks.

To balance these forces according to real pigeons [15, 24], we implemented a weighted sum (Eq 2) aiming for a strong influence of alignment ($w_a$), medium influence of avoidance ($w_s$) and acceleration-attraction ($w_{ca}$), and weak influence of centroid-attraction ($w_{ct}$). The exact values of these weights are established during calibration (Table 1). The total coordination force is calculated by:

$$\vec{\psi}_i^{coord} = w_s \hat{\psi}_i^s + w_a \hat{\psi}_i^a + w_{ct} \hat{\psi}_i^{ct} + w_{ca} \hat{\psi}_i^{ca} + \vec{\psi}_i^w \quad \text{Coordination force} \tag{2}$$

**Escape motion.** According to the empirical findings, the heading of the robotic falcon had a larger effect on the turn-away motion of pigeons than its position [24]. Hence, our pigeon-agents avoid the predator based on an escape pseudo-force ($\vec{\psi}_i^p$) that turns them away from the predator's heading. The magnitude of this force ($w_p$) is constant and independent from each agent's distance to the predator:

$$\vec{\psi}_i^p = w_p \, (sgn(\hat{h}_i \cdot \ \hat{h}_p) \, \hat{h}_i^\perp) \qquad \text{Predator\ avoidance} \qquad (3)$$

where $\hat{h}_p$ is the heading of the predator-agent, and $\hat{h}_i \cdot \hat{h}_p$ its dot product with a pigeon-agent's heading. Pigeon-agents do not sense the predator's position and thus have no information about their distance to it.

**The predator.** We model our predator-agents to resemble the motion of the robotic falcon [24] in order for our results to be comparable with the empirical data. Thus, hunting in our model is based on direct pursuit [24, 31], a common strategy of attack by peregrine falcons (*Falco peregrinus*) on flocks [32]. Each simulation includes only one predator. A few seconds after the flock is formed, the predator is positioned at a given distance behind it ($d_h$). The predator-agent follows the position of the pigeon-agent closest to it, with the same speed as the prey (based on the scaling factor $\zeta_p$) from a bearing angle $\theta_p$ relative to the flock's heading. After some time ($T_h$), the predator-agent will speed up attacking its target (with a speed scaling from the target's speed, $\xi_p$) with a random error added to its motion. The predator's target is selected based on two alternative strategies: the target is the pigeon-agent that is closer to the predator at every time step during the attack ('chase' strategy) or the closest pigeon-agent at the beginning of the attack ('lock-on' strategy). After an attack ($T_a$), the predator is automatically re-positioned far away from the flock.

**Update and integration.** We use two different timescales for update and integration in our model, following previous biologically-relevant computational models of collective motion [12, 33]. During update steps ($t + \Delta t$), agents collect information from their environment and the pseudo-forces acting on them are recalculated. Since flock members in nature are asynchronous in their reactions, our pigeon-agents update their information asynchronously with the same frequency ($\Delta t$). This asynchronous update adds noise to the system and it is known to improve the resemblance of collective motion in agent-based simulations to real groups [3, 12, 33]. At each integration step ($dt$), the total force for each agent is composed from:

$$\vec{\psi}_i(t) = \vec{\psi}_i^{coord} + \vec{\psi}_i^p + w_f \vec{\psi}_i^f(t) \quad \text{Total\ force} \qquad (4)$$

where $w_f$ is the calibrated weight for cruise speed control, and $\vec{\psi}_i^{coord}$ and $\vec{\psi}_i^p$ are the pseudo-forces for coordination (Eq 2) and predator-avoidance (Eq 3) calculated at the last update step. Based on this force, the acceleration, velocity, and position of all agents is updated at each integration step according to Newton's laws of motion and using Euler's integration (specifically the midpoint method [34]). The new heading of each agent is then the normalized form of its new velocity:

$$\hat{h}_i = \frac{\vec{v}_i}{||\vec{v}_i||} \quad \text{Agent's\ heading} \qquad (5)$$

## Experiments

For our main analysis, to test the effect of predator-prey distance on the pigeons' escape frequency, we performed two types of experiments. In both experiments, the predator is using the 'chase' strategy. First, we run simulations in which prey is not reacting to the predator.

These simulated data were used as control. In the second experiment, pigeon-agents avoid the predator-agent without accounting for its distance to them (as described in Eq 3). We ran 1000 simulations (hunting cycles) for each experiment. Across simulations, we varied only the direction of the predator's attack (from the left, the middle, and the right side behind the flock relative to the flight direction, $\theta_p$) and the flock size (8, 10, 27, and 34 individuals, as in the field experiment of homing pigeons [24]). Note that since we do not model 'catches' of prey by the predator, flock size is constant during each simulation. For our analysis, we combined the results of all simulations into one dataset per experiment.

To test the effect of the predator's strategy, we repeated our main experiments using (1) the 'lock-on' strategy and (2) reverted versions of the two strategies. Specifically, the predator was placed automatically in close proximity to the flock (5 m) and given a speed lower than its targets' speed; this resulted in the predator performing approximately the opposite motion of a real attack. We performed this 'reverted' experiments to test effects of hysteresis [5] during a collective escape.

## Analysis

The first seconds of each simulated flight are discarded to avoid effects of initial conditions. Similarly, the taking-off part of the real trajectories was excluded from our analysis. For the analysis on the predator-prey distance, data of individuals that are more than 60 meters away from the predator are also discarded, as in the empirical study with the RobotFalcon [24].

**Defining a flock of pigeons.** For both empirical and simulated data, we extracted the time-series of individual speed and nearest neighbor distance. We further estimated the bearing angle between every focal individual and each one of its neighbors based on its heading and the line connecting their positions. The shape of each flock was calculated based on the minimum volume bounding-box method [12, 35]. Specifically, we drew a bounding box of minimum area that included all flock members' positions, with axes aligned to the output of a principal component analysis of their coordinates (method used in [12]). To use as a proxy of flock shape, we estimated the angle between the shortest side (dimension) of this box and the heading vector of the flock (average of all flock members) [12]. The closer this angle is to 0 degrees, the more perfectly 'wide' the flock is. Values close to 90 degrees show an 'oblong' flock shape.

**Turning direction.** Members of both real and simulated flocks were categorized based on their turning motion relative to the headings of the flock and the predator (Fig 2A). Following the definitions of Sankey *et al.* (2021) [24], when the direction away from the predator's heading is also the direction away from the average heading of the flock, we will refer to the individual as being 'in-conflict' (between turning to either escape or align with the flock). In a non-conflict scenario, an individual needs to turn towards the flock's heading in order to escape. According to these categories, an individual may actually turn as follows:

1. in a 'conflict' scenario:

   a. towards the flock and the predator (potentially staying in the flock and risking getting caught),

   b. away from the flock and the predator (potentially splitting-off and escaping);

2. in a non-conflict scenario:

   a. towards the flock and away from the predator (potentially staying in the flock and escaping),

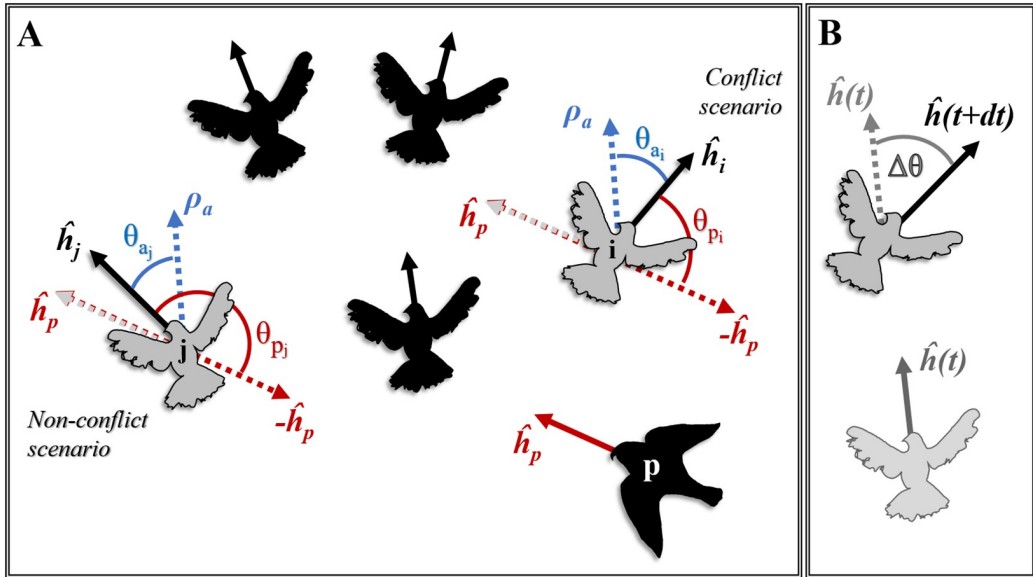

**Fig 2. Definition of turning direction.** (A) Prey individuals *i* and *j* are members of a flock with average heading $\rho_\alpha$ and face different conditions to escape the predator *p*. The headings (unit vectors) of the individuals are represented by $\hat{h}$. The red angles $\theta_p$ are the turns away from the heading of the predator and the blue angles $\theta_a$, the turns towards the heading of the flock. Based on the relative heading of individuals *i* and *j* to the headings of the flock and the predator, *j* needs to escape while aligning with its flockmates, while *i* is in-conflict, since it needs to turn away from the flock in order to escape. (B) The change of heading of an individual between consecutive time steps (*dt*) is represented by $\Delta\theta$. Its sign shows whether the individual turned towards the flock (same sign with $\theta_a$), away from the predator (same sign with $\theta_p$), towards neither or both (depending on its escape conditions).

 b. away from the flock and towards the predator (potentially splitting-off and risking getting caught).

We identified the turning direction of each flock member in respect to the predator by comparing the signs of a series of angles relative to the heading of the focal individual (Fig 2A). We let the sign of an angle between two vectors *a* and *b* be represented by the sign of their perpendicular dot product ($PerpDot(a, b) = \vec{a}^{\perp} \cdot \vec{b}$). When the *PerpDot* is positive, vector *a* is on the right of *b* (anticlockwise rotation of *a* is needed to overlap with *b*) and thus the angle $\theta_{ab}$ between them falls in the range [0˚, 180˚). When their *PerpDot* is negative, *a* is on the left of *b* (clockwise rotation is needed to overlap) and $\theta_{ab} \in [-180˚, 0˚)$. Thus, the sign of an angle between two vectors takes a value according to:

$$sgn(\theta_{ab}) = \begin{cases} 1, & \text{if } \vec{a}^{\perp} \cdot \ \vec{b} \geq 0, \\ -1, & \text{if } \vec{a}^{\perp} \cdot \ \vec{b} < 0. \end{cases} \tag{6}$$

Firstly, the direction 'towards the flock' is indicated by the sign of the angle $\theta_a$, the one between the focal individual's heading and the average heading of the flock ($\rho_\alpha$). Secondly, the direction away from the predator ($\theta_p$) is shown by the sign of the angle between the prey's heading and the vector ($-\hat{h}_p$) opposite to the predator's heading. Lastly, the direction of an individual's motion is estimated as the sign of the angle between its heading at consecutive integration time-steps ($\Delta\theta(t + dt)$, Fig 2B). After comparing these three directions, we categorized each individual's motion in the above mentioned categories (1a, 1b, 2a, 2b).

As in the analysis of Sankey *et al.* (2021) [24], we split our data into 10-meter clusters of predator-prey distance (from 0 to 60 m). We calculated the frequency of each turning category at each cluster across all real and simulated trajectories. We refer to the frequency of turns away from the predator (1b, 2a) as 'escape frequency'. Since we are interested in the emergence of a collective pattern, please note the distinction between escape frequency (the actual behavior of an individual, emerging from its interactions with its neighbors and the predator) and the underlying rule for predator-avoidance (the individual tendency to escape).

**Distance dependency.** We analyzed the simulated data focusing on how several variables scale with decreasing distance to the predator. Specifically, we calculated for each predator-prey distance cluster:

1. the angle ($\alpha_{pi}$) between the headings of the predator-agent ($\hat{h}_p$) and each prey-agent ($\hat{h}_i$) at each time step.

2. the frequency that pigeon-agents change their escape direction across all simulations:

$$v_c = \frac{1}{N\, T_{sim}} \sum_{i \in N} n_i^s \tag{7}$$

where $N$ is the total number of pigeon-agents in all simulations, $T_{sim}$ the total simulation time, and $n_i^s$ the number of occurrences that the escape direction of agent $i$ changed between time steps ($\theta_{p_i}(t) \neq \theta_{p_i}(t+1)$).

3. the consensus in escape direction in each flock at every time step:

$$C_f^{esc}(t) = |\frac{1}{N_f} \sum_{i \in N_f} sgn(\alpha_{pi}(t))| \tag{8}$$

where $N_f$ is the number of individuals in the flock $f$ and $sgn(\alpha_{pi}(t))$ the sign of the angle between the headings of the predator and individual $i$ at time $t$. Values close to 1 show that most flock members have the same escape direction, whereas 0 indicates that half of the flock needs to turn to the right to escape and the other half to the left.

**Self-organized dynamics.** To inspect the interplay of coordination and predator-avoidance during collective escape in our model, we examined the effect of alignment and centroid-attraction on the motion of each individual. Specifically, we collected the weighted steering forces ($\hat{\psi}_i^a$, $\hat{\psi}_i^{ct}$, Eq 2) of each flock-member in its own coordinate system:

$$F_f^\psi(t) = \{w_a \hat{\psi}_1^a,\ w_{ct} \hat{\psi}_1^{ct},\ \dots,\ w_a \hat{\psi}_{N_f}^a,\ w_{ct} \hat{\psi}_{N_f}^{ct}\} \tag{9}$$

where $F_f^\psi(t)$ is the set of the coordination forces acting on all $N_f$ focal individuals of flock $f$. We used these sets to create density maps of coordination effect during a predator's attack on a flock.

## Software

Our computational model was built in C++ 17. Graphics rendering was implemented in OpenGL [36]. The calculations of bearing angle, flock shape, and predator-prey distance, and the determination of conflict scenarios for the simulated data were performed in C++. All other analyses of empirical and simulated data were performed in R (version 3.6) [37]. All result plots were made in 'ggplot2' [38].

## Results

### Characteristics of a flock of homing pigeons

We analyzed tracks of homing pigeons in flocks, initiating their homing flight or being attacked by a robotic falcon ($N = 43$). To characterize what comprises a flock of pigeons, we constructed for each flock the distribution of individual speed, nearest neighbor distance, and calculated for each focal individual the relative position of all its neighbors (distance and bearing angle) (Fig 3A1, 3B1 and 3C1). We found large differences in these distributions among flocks (S2 Fig). The bearing angle to each nearest neighbor (S3 Fig) is uniform distributed (Kolmogorov-Smirnov test, comparison with a uniform distribution, D = 0.08,

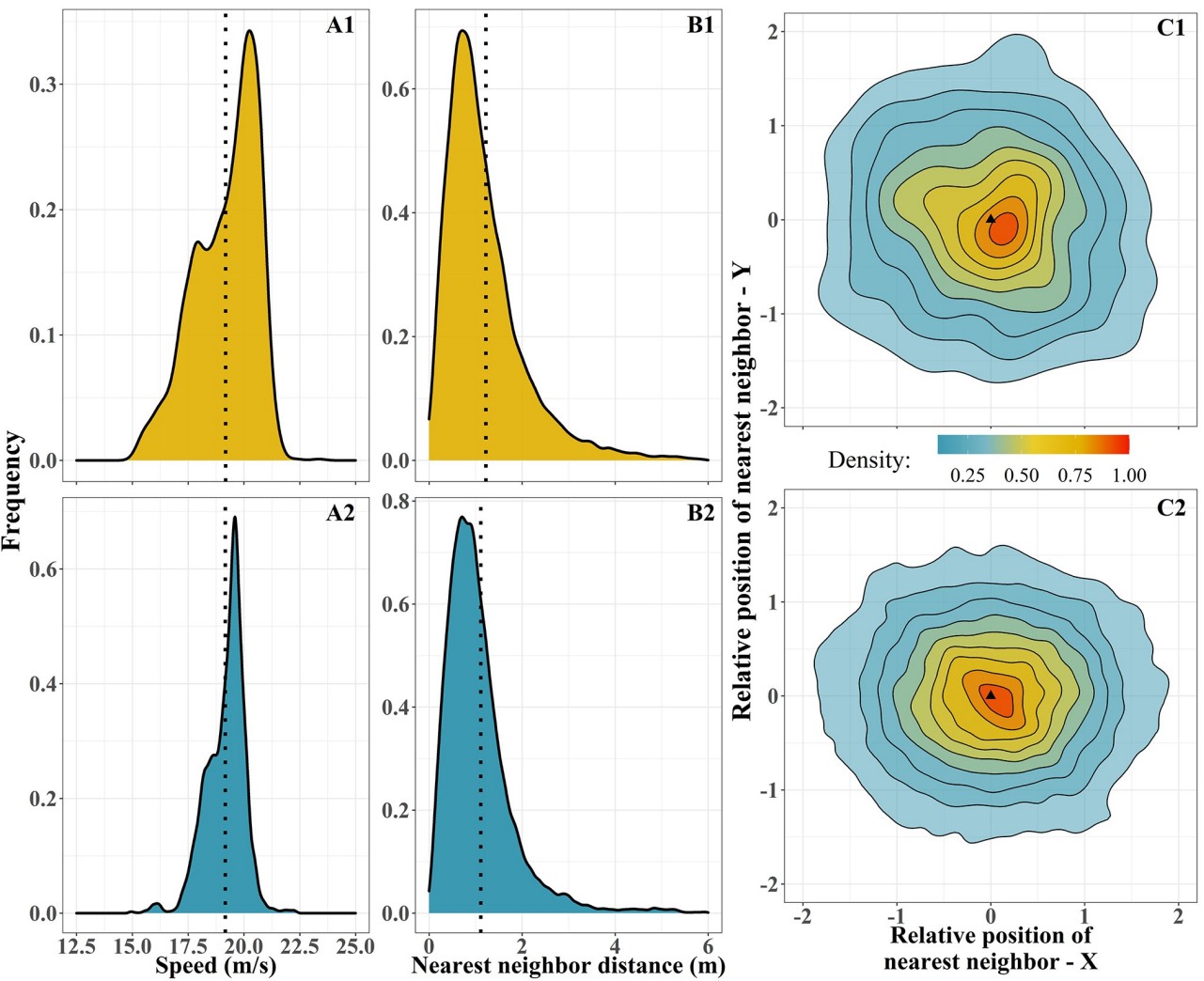

**Fig 3. Comparison of real (top row) and simulated (bottom row) flocks of 8 pigeons.** The shown data are representative of the majority of distributions; for a detailed comparison across more flocks see S1 Table. (A-B) The vertical dotted lines show the mean of each distribution. (A) The distributions of individual speed throughout a flight of real pigeons (A1) and a simulation (A2). (B) The distributions of nearest-neighbor distance throughout a flight (B1) and a simulation (B2). (C) Density of the nearest-neighbor positions in the coordinate system of each focal individual, based on the bearing angle and distance to the nearest neighbor ($m$), estimated for a real flight (C1) and a simulation (C2). The area of highest density is expected to be around the mode of the distribution of nearest neighbor distance. The triangle represents the position of the focal individual heading to the north direction.

p-value = 0.58). The shape of each flock varies from oblong to elongated, according to our calculation of the angle between the flock's heading and the shortest axis of the minimum-area bounding box surrounding all members (with an average angle of 41 ± 23.5 degrees). In total, in a flock of homing pigeons, an individual flies with an average speed of 18 *m/s* and has its nearest neighbor positioned anywhere around it (bearing angle from -180 to 180 degrees) at a distance of 1.3 ± 1.8 meters (mean and standard deviation).

## Simulated flocks of pigeons

Before inspecting our hypothesis, we established whether the behavioral rules of our model result in pigeon-like flocks [17]. To develop our model, we adjusted the relative importance of the coordination rules (alignment, attraction, avoidance) among individuals according to empirical data of pigeons. For each pigeon-agent, we modeled strong alignment with the average heading of its 7 closest neighbors, and weak attraction to their center of mass. As to attraction, we also included an accelerating mechanism, as found by Pettit *et al.* (2013) [15]. While individuals accelerate to stay close to their flockmates, an additional force drags them back to their personalized cruise speed [21].

We measured the flock characteristics described above (distribution of speed, nearest neighbor distance and relative position of neighbors) in our simulated tracks. Due to the large variability in measurements among real flocks, the cumulative distribution across flights is not representative of individual flocks (S2 Fig). Hence, we compared the simulated data with tracks of a single flock and validated that our computational model closely resembles flocks of pigeons (Fig 3 and S1 Table).

## Turning-direction frequency under attack

We model pigeon-agents to avoid the heading of the predator rather than its position, according to the findings of Sankey *et al.* (2021) [24]. Specifically, we added a force perpendicular (in ±90 deg angle) to the heading of each agent, pushing them to turn away from the predator (while also coordinating with their neighbors). The strength of this force is constant and independent of the predator-prey distance. Only its direction (the forces' sign) changes during an attack, based on the relative direction of the prey's and predator's headings. Our simulations show a pattern very similar to the one of the empirical data: the closer the predator, the higher the frequency of escape is (Chi-Square for Trend test, X-squared = 10.3, p-value = 0.001, Fig 4B), independent of our predator-agent's strategy ('lock-on' and 'chase', S4 Fig).

This shows that in our model, the pattern of collective escape that scales with predator-prey distance emerges from an individual behavior that does not take their distance to the predator into account (distance-independent avoidance). In simulations where pigeon-agents do not react to the predator (control), the frequency of escape remains constant (Chi-Square for Trend test, $x^2$ = 0.03, p-value = .87, Fig 4C), as expected.

## Parameters scaling with predator-prey distance

We examined several aspects of our system to identify why pigeons turn more frequently away from the predator when they are closer to it (Fig 4). We focused on measurements that differ between simulations with and without predator-avoidance, to eliminate effects of mechanisms that are irrelevant to predation. When pigeon-agents do not react to the predator, the values of the chosen measurements show little or no variation with predator-prey distance (Fig 5A1, 5B1 and 5C1).

Fig 5A2 visualizes the progression of a collective turn: the closer the predator, the larger the angle between its heading and the preys' headings. By turning away from the heading of the

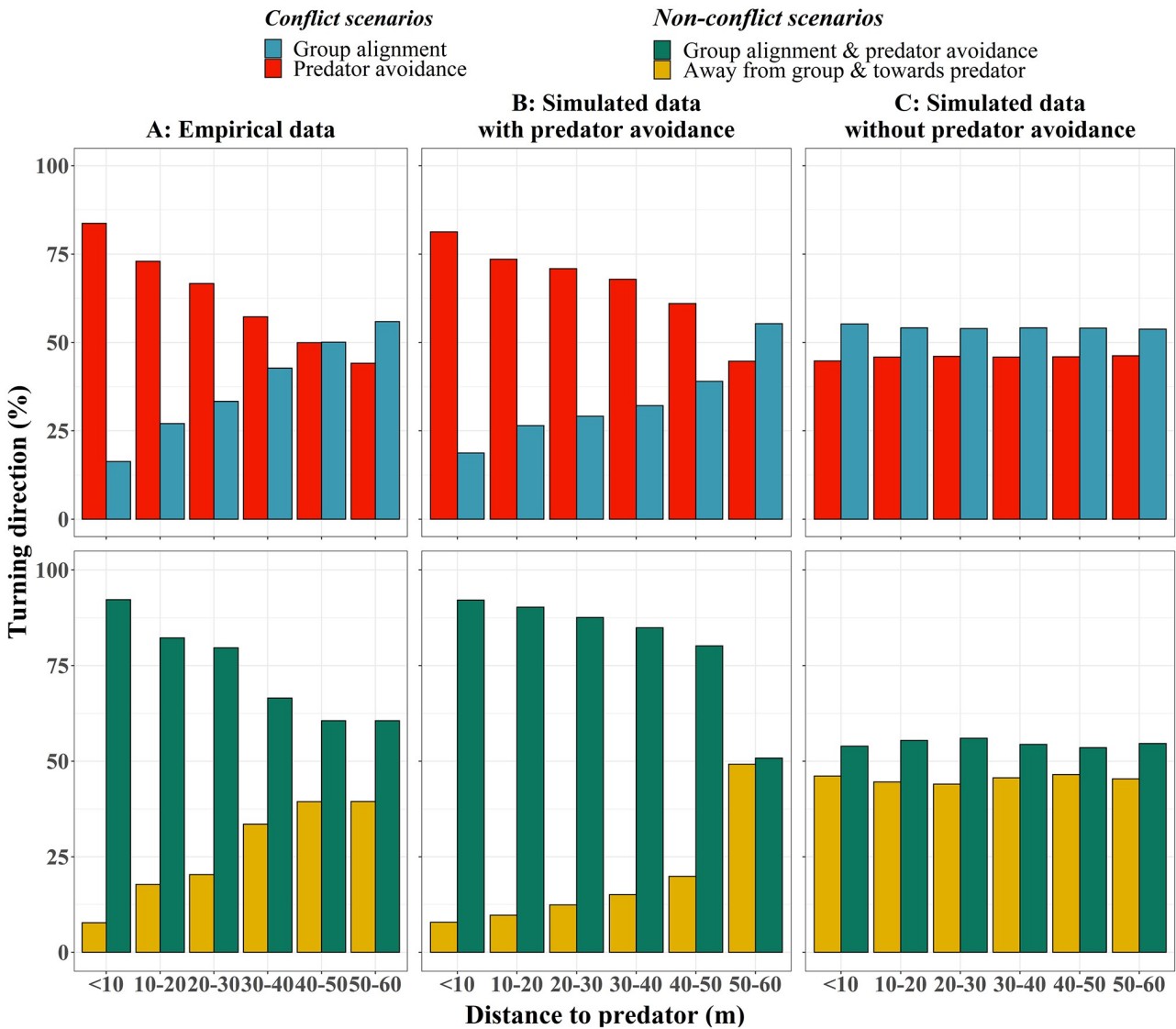

**Fig 4. Turning direction frequencies of flock members.** The percentage of turns towards the four turning directions at consecutive time-steps (for individuals under 'conflict' and 'non-conflict' scenarios) as a function of distance between them and the predator, across the empirical data and two simulation experiments. (A) Empirical data of Sankey *et al.* (2021) [24]. (B) Simulated data with predator avoidance that is independent of the distance between the predator and the prey individuals (modeled as in Fig 1). (C) Data from control simulations where the prey does not react to the predator.

predator, agents reinforce the predator-avoidance force to be towards the same direction (relative to their headings) across consecutive time steps. In detail, a few individuals start turning first, influencing through alignment and attraction their neighbors to turn as well. Through this turning motion, the angle between flock and predator increases. This makes the escape direction to be more robust, since small deviations in heading of pigeon-agents do not alter their escape direction (in other words, their heading is still on the right or the left relative to the predator's heading). As a result, the frequency with which the predator-avoidance force changes direction decreases as the predator gets closer (Fig 5B2).

When the predator is very close, aiming to intersect the flock's trajectory (for an example see Fig 6A5), the escape direction across the flock is almost constant, since the flock is already

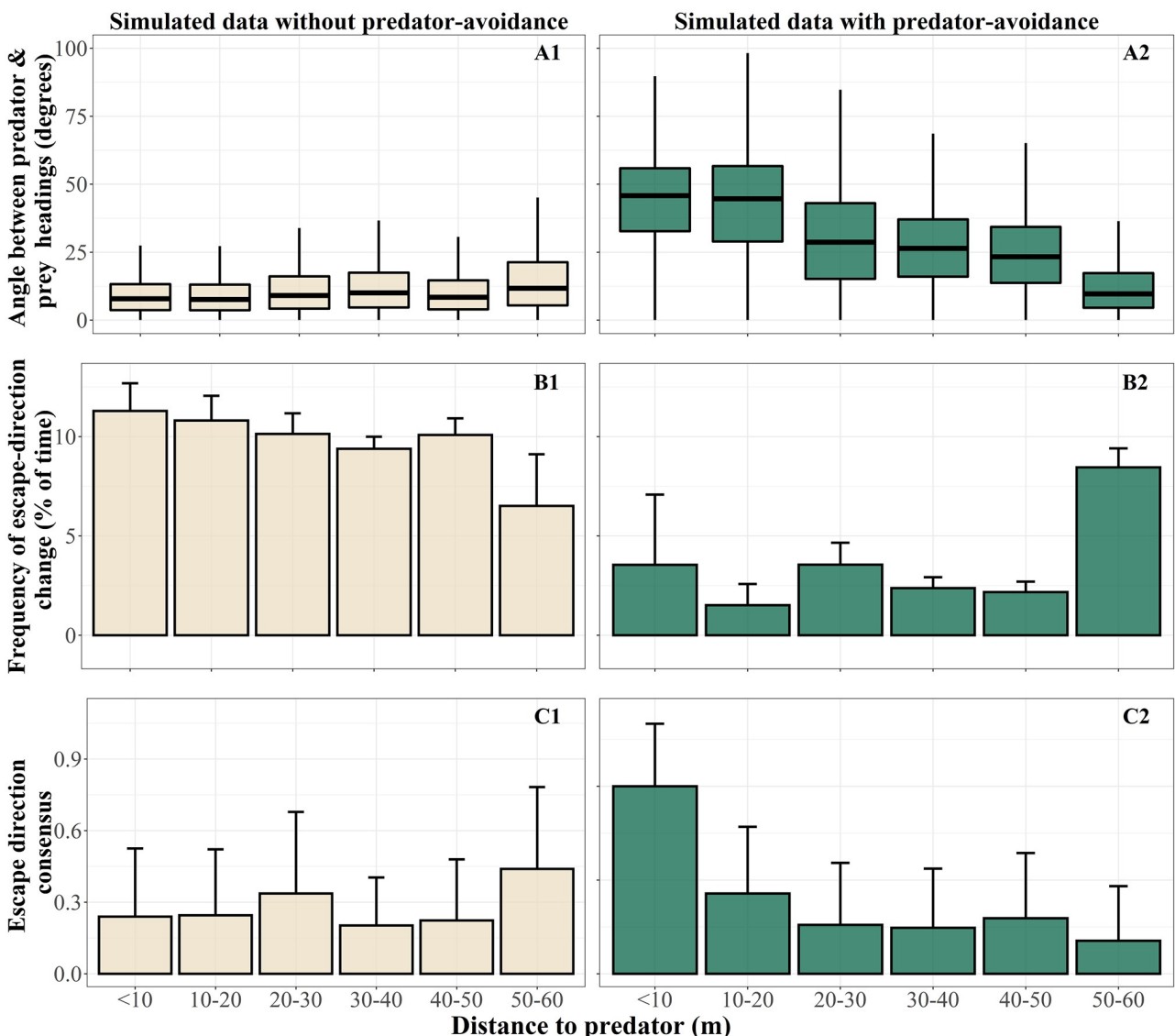

**Fig 5. Distance-dependency in simulated flocks with and without predator avoidance.** (A) The angle of the flock members' headings relative to the predator's at each distance-cluster. The boxes include the 50% of the distribution and the horizontal line shows its median. When pigeon-agents react to the predator, the measurement increases showing that the flock is turning away from the predator. (B) The frequency that the escape-direction of each flock-member is changing direction (Eq 7). The height of each bar shows the mean value of all individuals per distance cluster and the error bar shows one standard deviation above this mean. The escape direction remains more stable when the flock is turning away from the predator. (C) Consensus in escape direction across a flock at each sampling point (Eq 8). More flock members have the same escape direction closer to the predator in simulations with predator-avoidance.

performing a collective turn and the angle between the headings of the flock members and the predator is large. Thus, the escape direction has the same sign (to the left or to the right) for all individuals. Through this mechanism, we see the consensus in escape direction across the group increasing with decreasing distance to the predator (Fig 5C2). Based on these components, in the next section we aim to explain how the escape frequency of group members increases through self-organization when the predator gets closer.

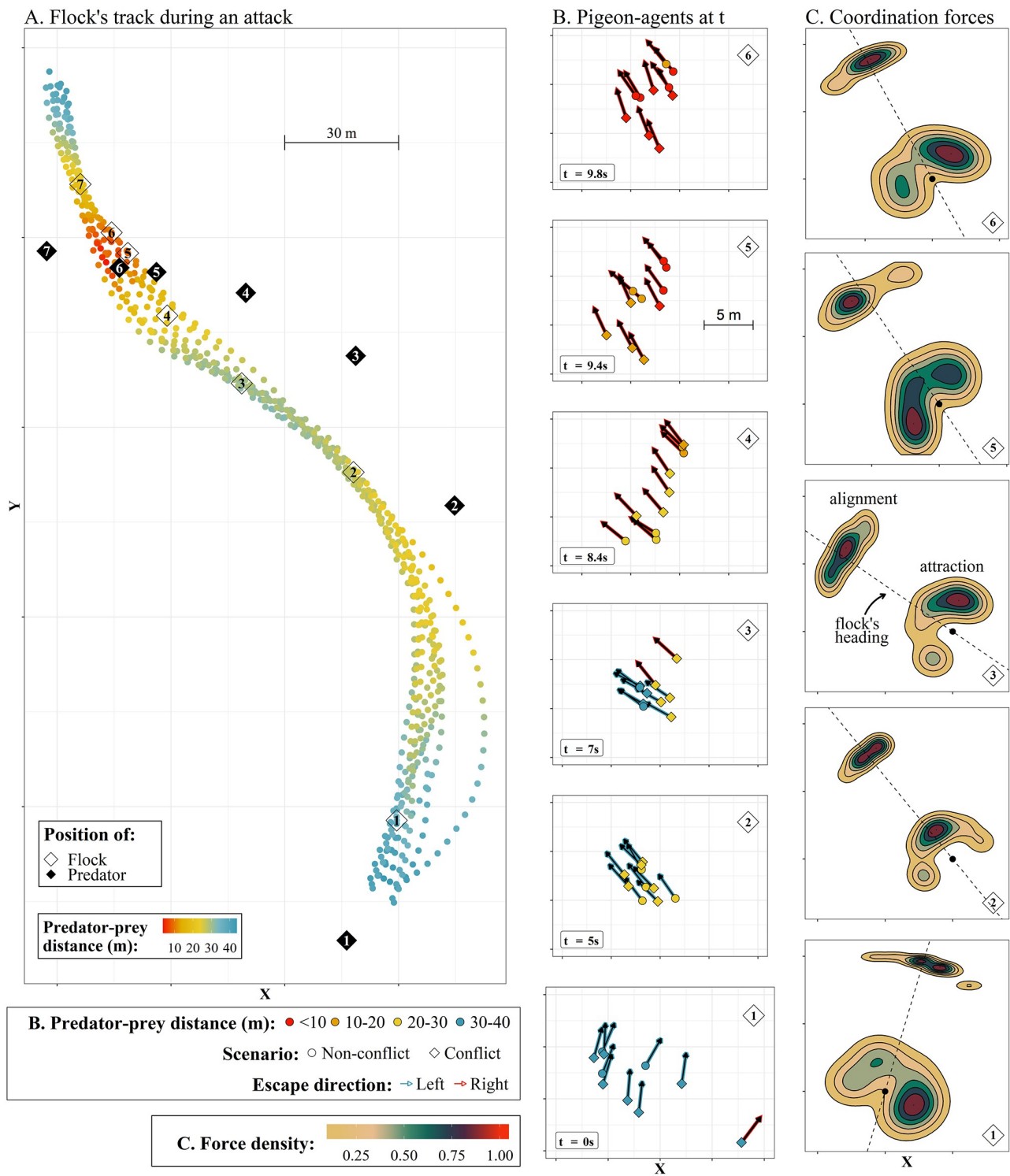

**Fig 6. Progression of a collective escape.** (A) The tracks of a simulated flock of 10 pigeons under attack. For simplicity, we present the part of the track when the predator is within 40 meters distance from the flock (excluding larger distances shown in previous plots). The points represent the position of single pigeons per time step of 0.2 seconds. The filled black rhombi show the position of the predator at 7 discrete time points. The numbers represent the link between the position of the predator and prey in time. (B) Positions of the pigeon-agents at time points 1 to 6 (0 to 9.8 seconds). Their color shows the distance to the predator of each individual (according to the clusters of Fig 4). Arrows represent the heading of each agent. The shadow of the arrows shows the escape direction of each agent at that time point. (C) The effect of centroid-attraction and alignment forces across the flock during the respective time points (density map of $F_f^\psi(t)$, Eq 9). The point represents the position of all individuals in their local reference frame and the dotted line shows the average heading of the flock at that time point.

## The role of self-organization

By combining our results, we form a theory that this pattern emerges by self-organization from the coordination among individuals during collective turning.

From the measurements that scale with predator-prey distance (Fig 5), we concluded that the progression of the collective turn is necessary for this pattern to emerge. Hysteresis [5] may play a role in the identified pattern on turning direction; the state of the flock at a specific distance to the predator has an effect on the pattern of the next state (at a closer distance). This was supported by our additional experiment where we positioned the predator close to the flock at initialization with a speed lower than the flock's. As the predator is drifting backwards (in an almost opposite motion to the one of its normal attack), the distance-dependent pattern does not arise (S4 Fig). Specifically, the turning-direction frequency does not differ among the clusters (<10 to 50 meters) of predator-prey distance (*p-value* = 0.171, $X^2$ = 1.875), in contrast to the effect we see in our main simulations (*p-value* < 0.005, $X^2$ = 10.321). In this scenario, the group doesn't have enough time to reach consensus over its escape direction.

To take a closer look at coordination among flock members during a predator attack, we inspected the effect of the coordination forces of centroid-attraction and alignment on the turning motion of our pigeon-agents throughout the simulations. In Fig 6, we show the details of coordination and the resulted collective motion during an attack: two consecutive collective turns of a flock as a reaction to the predator (Fig 6A). Note that the consecutive turns emerge in the majority of pursuits in our simulations. They probably relate to the predator overshooting when initially approaching the flock (a behavior that is theoretically expected when prey avoids a predator's heading [24, 39]). At intermediate distances (20–30 meters) to the predator, increased consensus in escape direction (Fig 5C2) is causing the alignment and cohesion force to both pull individuals away from the predator into a turn. We describe the process below, referring to the simulated data of the single attack shown in Fig 6. We classify the agents that are at the side edges of the flock, based on their relative position to the flock's direction of turning, into 'inner-edge' (e.g. on the left side of the flock in a left escape-turn) and 'outer-edge' (on the opposite side).

**Predator's behavior.** As the predator is approaching, the flock is continuously turning away from its heading. We observe the predator getting closer to the flock after a change in the direction of its attack (as seen in Fig 6A1 and 6A2), by crossing the flock's path from the side (Fig 6A5–6A7).

**Flock shape.** The shape of the flock has an important effect on the dynamics of a collective turn. In the beginning of a turn, the flock has a relatively wide shape (Fig 6B1 and 6B4). Exiting a turn, the flock becomes oblong (Fig 6B2). In this shape, the centroid-attraction is in the direction of the flock's heading (Fig 6C2). Simultaneously, the alignment-force acts also closely around the flock's heading (Fig 6C2); the flock is very polarized. When the predator catches up and approaches from the opposite direction, the inner-edge individuals switch their escape direction first (Fig 6B3). By starting to turn, they change the shape of the flock to wide (Fig 6B3 and 6B4) and move the effect of coordination forces (center of the flock and average heading) towards the escape direction (Fig 6C3). With the coordination forces acting in the same direction as escape, the whole flock enters the turn (Fig 6B4 and 6B5). When this happens, the center of the flock re-positions to the opposite direction of escape (Fig 6C5). The individuals that have started the turn (inner-edge) are now more attracted to the non-escape direction (Fig 6C5). Outer-edge individuals (on the opposite side of the flock's center) are still moving inwards (Fig 6B5). The flock is becoming more oblong and approaches the exit of the turn (Fig 6B6), where alignment and coherence will again have a subtle effect on the individuals' turning (by acting around the flock's heading, Fig 6A7, as in Fig 6B2 and 6C2).

**Progression of turn.** At one time-point, flock members may be in different distance-clusters to the predator (Fig 6B3–6B6). The individuals that are closer to it, at the inner-edge of the group, will establish the common escape direction first (Fig 6B3, red shaded arrows). These individuals ('inner edge') make the flock shape wider and move the center of the flock and the average alignment towards the escape direction, initiating the turn (Fig 6A3). Since the turn propagates from the inner-edge individuals, the outer-edge individuals (on the outside of the flock) have a delay in starting the escape turn (Fig 6A4 and 6B4). At this point, outer-edge individuals are not in conflict, since their escape direction matches the average alignment of the flock. Further in the progression of the turn, with more individuals turning to escape, the shape of the flock becomes more oblong. For the initiators of the turn (inner-edge), the center of the flock re-positions towards the non-escape direction (Fig 6B5 and 6C5). For the outer-edge individuals, the center of the flock is in the direction of escape (Fig 6B5 and 6C5). By sharply turning towards the escape direction to catch up with the flock that started turning earlier (Fig 6A4 and 6A5), they are in-conflict, since for them alignment acts in the opposite direction (Fig 6B5, 6B6, 6C5 and 6C6). Reaching the exit of the turn, alignment acts close to individuals' headings and attraction switches back, in accordance with the escape direction for most flock members (Fig 6C6). This makes the flock to continue the turn until the elongated shape where attraction and alignment are around the agents' headings (Fig 6A7, as in Fig 6A2, 6B2 and 6C2).

**The role of 'in-conflict' individuals.** During an escape turn of the flock, different individuals are 'in-conflict' between keeping up with the flock's heading or avoiding the predator. At the beginning, inner-edge individuals are in-conflict, resulting in them initiating the turn (Fig 6B3). When the predator gets closer, and further in the progression of the turn, the outer-edge individuals (the last to start turning) are in-conflict. This results in cohesion acting in accordance to escape. The observed increase in escape with predator-prey distance may be caused by alignment and attraction acting in opposite directions for individuals in-conflict close to the predator (Fig 7). By the direction change of the coordination forces and the change in the conditions that each individual is under (depending on their relative positions in the flock) during the turn, the flock manages to stay together, while different parts are 'in-conflict'.

**Mechanism's summary.** In total, at the beginning of the turn, in-conflict individuals are the ones initiating the turn due to the predator avoidance force. With the turn progressing, the flock shape becomes wide and by the in-conflict individuals having the center of the flock on their escape direction, cohesion acts in favor of escape. Towards the end of the turn, the initiators are moving more inwards, while the outer-edge individuals are the ones in-conflict, catching up with the escape turn.

In other words, when in-conflict at large distances (the beginning of the turn), only the predator-avoidance force acts in accordance with the escape direction, while the flock is oblong and polarized (thus coordination forces have small effect on turning). When the predator is closer (in the progression of the turn), attraction and alignment are in opposition. Attraction to the center of the flock is in accordance with the escape direction. Since the flock has reached consensus concerning the escape direction, individuals turn more towards the escape and attraction direction instead of towards the alignment direction (their neighbors' heading).

## Discussion

Computational models based on self-organization have helped to unravel what behavioral rules underlie collective phenomena in group living organisms [1, 5, 12, 19]. Specifically, these models show that many collective patterns emerge from the interactions among group

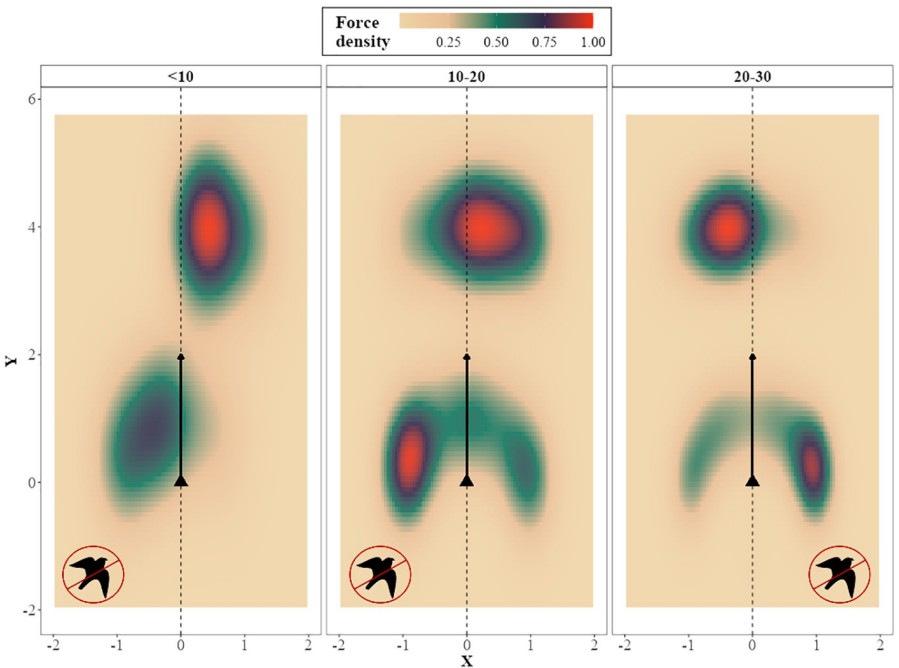

**Fig 7. Effect of coordination forces on 'in-conflict' flock members within 30 meters to the predator in *HoPE*.** The density of turning-attraction (low center) and alignment (high center) forces (Eq 9) acting on the coordinate system of pigeon-agents that are in-conflict during the pursuit sequence shown in Fig 6. The triangle represents the position of each focal individual and the dotted line and arrow its heading. The predator sign represents the turning direction of predator avoidance. Alignment and centroid-attraction have opposite effects on turning direction relative to the agents' headings and predator-avoidance is mostly in accordance with the centroid-attraction direction.

members, rather than being predefined in behavioral rules at the individual level [7, 40]. In the present study, we use such a computational model to explain how pigeons in flocks increase their frequency of turning away from the predator, the closer they get to it. This distance-dependent collective pattern was initially hypothesized to reflect that, when prey were closer to the predator, they increasingly prioritized to turn away from the predator rather than align with their flock's heading [24]. In our model, this pattern of collective escape emerges without individuals having a distance-dependent rule for avoiding the predator. Specifically, it is a side-effect of interactions among individuals as the predator is getting closer, because the direction towards which they should turn to avoid the predator is being reinforced among flock members. Thus, our results show that prioritizing escape over coordination, as suggested empirically, is not necessary for group members to increase their frequency of turning away from the predator when the predator gets closer.

An important difference in our conclusion from that of the empirical study is that flock members do not need to consider their distance to the predator in order to escape collectively. In reality, this mechanism may spare prey the cognitive effort [41, 42] of continuously keeping track of the predator's position during coordinated motion. Which individuals start turning away from the predator first, happens spontaneously, similarly to the initiation of collective turns measured in flocks in nature [43, 44]. They thus affect the escape motion of the rest of the group via other group members: the neighbors of an initiator turn towards its position and heading, resulting in the neighbors of its neighbors also following, until all group members (that are not direct neighbors of an initiator) are turning away from the predator. In other words, information concerning the change of heading propagates through the group [45]. Our

model shows that this reinforcement increases the growing consensus over the direction of escape the closer the predator gets, through a quorum-like response [46]. This information transfer of changes in heading in our flocks seems central to collective escape, as it is also found across different species [47, 48].

In more detail, the distance-dependent pattern emerged through self-organization from a combination of processes. First, a distance-independent tendency to turn away from the predator, leading the group into a collective turn. Secondly, the fact that centroid attraction acts in the direction of escape that is opposite to that of alignment when the flock is closer to the predator. Thirdly, through short-term hysteresis [5]: during the progression of a collective turn, as the predator approaches, the past state of the flock (shape and relative positions of flock members) affects its next state. Both flock shape and internal structure relate to what each individual experiences in terms of predator threat (angle of attack and escape direction) and social coordination (deviation from the flock's heading and relative position to the centroid), and affect the propagation of information concerning changes in heading through the group [31]. To our knowledge, the effect of hysteresis is new in the context of collective escape.

The switch by the majority of the group in our study between turning away from the predator and aligning with the group (Fig 4) differs from studies on phase transitions in collective decision-making [49–52] in a number of ways. First, in our model there is neither a proportion of individuals that is 'informed' to a pre-designed preferred direction of escape (like 'goal-directed' leaders) nor that is more prone to escape than their flockmates [49, 52]. Secondly, the alternative directions in our study ("towards the flock's heading" and "away from the predator") are not global or fixed in space, they depend for each individual on its specific local environment. Thus, initiators of collective turns emerge based on their position and heading relative to the group and the predator and individuals can instantaneously change their turning direction at any time. For instance, the individuals that started turning away from the predator first, may turn towards it further in the progression of the collective turn, while the rest of the flock is turning towards the initial escape direction. Moreover, our two directions are mutually exclusive (individuals cannot move towards the average of the two [52]) and the group always collectively increases their predator avoidance rather than their alignment when closer to the predator (i.e. the direction towards the group heading is never collectively selected). To understand the mechanisms underlying collective decision-making across contexts, future research should investigate these dissimilarities and the differences between the role of leaders, goal-directed individuals and initiators [53].

Predation is known to affect the coordination within groups of prey [31, 54, 55]. An example is a decrease in the minimum separation distance [54] and an increase in the number of interacting neighbors [55] in the presence of a predator. Such changes lead to increased group density in fish [54], a pattern not seen in pigeons under attack by a predator [24]. Since collisions are a large threat for birds in flight, a decrease in nearest neighbor distance or an increase in centroid-attraction may enhance the danger of colliding with each other. According to our results, a stronger tendency to align with flock members than to turn towards the flock's center increases the prey's escape frequency at shorter distance to the predator while retaining flock cohesion during collective turns. We thus hypothesize that for small flocks that turn away from their predator, increased alignment rather than decreased group density enhances their chances of survival.

Whether prey escape by considering the predator's position or its heading is unclear. In fish schools and insect swarms, individuals are supposed to avoid the position of the predator [48, 56]. Homing pigeons instead were observed to turn away from its heading [24]. This heading-avoidance seems to indicate avoidance of where the predator will be in the future. Such anticipation-based strategies are known to be used by predators to catch prey [57]. In our

simulations, we observed that with heading-avoidance a common escape direction is enforced among group-members supporting group cohesion during collective escape. In a previous model of fish schools, we see that when individuals turn away from the position of the predator, the group splits more frequently when the predator gets closer [7]. The adaptivity of these avoidance strategies may depend on the species and ecological context. For instance, if prey is very maneuverable (i.e. fish and insects rather than birds [54, 58–61]) or subject to surprise attacks by their predator [62, 63], avoidance of the position of the predator may be more favorable.

In our model, the pattern of increasing escape frequency when the predator gets closer is robust across variations of our experiments, where pigeon-agents avoid the position of the predator (S5 Fig, even though this is not a behavior of pigeons as identified by [24]). This suggests that the collective pattern reflects a consensus in escape direction, and the collective decision to turn, irrelevant of some specifics of the underlying avoidance rule. For ease of explaining the emerging mechanism, we focus on a representative track from our simulations in Figs 6 and 7 that shows changes in the attack direction of the predator (possibly due to overshooting [24, 39]) and consecutive collective turns of a flock. Alternative predator-avoidance rules (e.g. avoiding both the heading and position of a predator) or other elements affecting the pursuit pattern (e.g. speed difference between prey and predator) may be interesting points for future theoretical research.

The strategy of attack by a predator may also affect the pattern of collective escape. We tested two attack strategies in the model, both based on direct pursuit: the predator follows and attacks the flock from behind [24, 31, 32, 64] by either locking-on the closest prey at the beginning of the attack or by chasing the closest prey at each time point. We showed that our results for the two strategies are similar (S4 Fig). These strategies have been previously used in computational models of collective escape [7, 40] and are similar to the pursuit performed by the RobotFalcon [24] and peregrine falcons (*Falco peregrinus*) in nature [32]. Since real predators also have alternative attack strategies (based, for instance, on anticipation [57, 65]), future research may focus on the effect of the predator's strategy and the specifics of its motion on the prey's escape behavior [61, 66, 67]. Given the evidence that flocks recognize the RobotFalcon as a real predator [24, 25], remotely-controlled predators can be a valuable tool to progress in this direction.

Our findings are relevant for collective escape by homing pigeons, given that flocks in our model resemble those of real pigeons not only in their behavioral rules [15, 21, 24], but also in their emergent properties (e.g. distributions of speed and nearest neighbor distance) [17]. Note however that our model is a caricature of reality and does not aim to capture all the variability found in the empirical data (see [17] for details on balancing data complexity and model simplicity in pattern-oriented modeling). Furthermore, we built our model in two dimensions given that: (a) the analysis of the empirical data by Sankey *et al.* (2021) [24] was done only using the x and y coordinates of the GPS trajectories [24], (b) pigeon flocks are generally flat (with little expansion in altitude) [68], and (c) collective turns are often on a plane, even in 3-dimensional flocks of starlings [45]. Recent work on the collective escape of fish schools further supports the robustness of conclusions made by analyzing only 2 out of the 3-dimensions of groups under attack [69].

Similarly to previous species-specific models of collective behavior of different species [3, 12, 20], our model can be further used to study other aspects of flocking in pigeons, be extended to investigate evolutionary dynamics of collective escape [70], or be adjusted to other bird species. The increasing availability of quantitative data of collective behavior can, and should, further support the development of models around specific species to help interpret empirical findings.

## Supporting information

**S1 Table. Comparison of summary statistics of speed and nearest neighbor distance between 20 real and 20 simulated flocks.** Outliers (above the top 99% of the distributions) have been removed. Since in the real flocks individuals are still taking off at the beginning of the pursuit, we further removed the bottom 1% of the speed distributions of the empirical data. Our simulated data fall within the range of real flocks.
(PDF)

**S1 Fig. Testing the accuracy of the topological-range estimation method of Sankey *et al*. (2021) [24].** Their method is based on simple linear models between, on one hand, the turn that each individual performs during consecutive sampling points and, on the other hand, the turning angles for centroid-attraction and alignment. Angles based on all possible topological ranges for each flock size were tested. The linear model with the most explanatory power was thought to include the 'real' topological range. For the exact method description see [24]. Given that this method is not well established, we tested its performance on our simulated datasets. Specifically, we applied this method on data from simulations in which we vary separately the topological range for alignment and centroid-attraction (from 1 to all neighbors) for three flock sizes. We run 5 repetitions of each simulation with all unique combinations of topological range for alignment and centroid-attraction per flock size. The deviation index shows the deviation of the topological estimate of the linear-models method from the real value of topological range as parameterized in the model (shown on the x-axis), divided by the maximum possible deviation for each topological range and flock size (n-2, e.g. the maximum deviation for a flock of 8 individuals is 6 neighbors, when the true value is 7 and the estimate is 1 or vice-versa, giving a deviation index of 1). Values close to 0 show a good performance of the linear-model method. The method seems to lose accuracy when agents align with many topological neighbors and when they are attracted to the centroid of a few. Each point shows the mean deviation index of all simulations with the respective topological range and the error bars the standard error.
(TIF)

**S2 Fig. Distributions of speed, nearest neighbor distance, and shape of flocks of homing pigeons.** Each histogram shows the distribution of one flock during a control flight (based on the data of Sankey *et al*. (2021) [24]). The bottom row shows the overall distribution across flights.
(TIF)

**S3 Fig. Distributions of bearing angle to nearest neighbor in flocks of homing pigeons.** The overlapping histograms show the distribution of one flock during a control flight (based on the data of Sankey *et al*. (2021) [24]). The bottom row shows the overall distribution across flights.
(TIF)

**S4 Fig. Turning direction from all alternative simulation experiments with different predator strategy.** The default strategy is the 'chase closest prey'.
(TIF)

**S5 Fig. Turning direction from the two alternative escape strategies of pigeon-agents.** The default strategy, supported by the empirical data [24], is avoidance of the predator's heading (A). The pattern of increased escape frequency of pigeon-agents at closer distance to the predator-agent holds for both strategies.
(TIF)

## Author Contributions

**Conceptualization:** Marina Papadopoulou, Hanno Hildenbrandt, Charlotte K. Hemelrijk.

**Data curation:** Marina Papadopoulou, Daniel W. E. Sankey.

**Formal analysis:** Marina Papadopoulou.

**Funding acquisition:** Steven J. Portugal, Charlotte K. Hemelrijk.

**Investigation:** Marina Papadopoulou.

**Methodology:** Marina Papadopoulou, Hanno Hildenbrandt, Charlotte K. Hemelrijk.

**Project administration:** Charlotte K. Hemelrijk.

**Resources:** Hanno Hildenbrandt, Daniel W. E. Sankey, Charlotte K. Hemelrijk.

**Software:** Marina Papadopoulou, Hanno Hildenbrandt.

**Supervision:** Charlotte K. Hemelrijk.

**Validation:** Marina Papadopoulou.

**Visualization:** Marina Papadopoulou.

**Writing – original draft:** Marina Papadopoulou.

**Writing – review & editing:** Hanno Hildenbrandt, Daniel W. E. Sankey, Steven J. Portugal, Charlotte K. Hemelrijk.

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
