## [Decision Letter · Decision Letter 0]

8 Sep 2021

Dear Mrs Papadopoulou,

Thank you very much for submitting your manuscript "Self-organization of collective escape in pigeon flocks" for consideration at PLOS Computational Biology.

As with all papers reviewed by the journal, your manuscript was reviewed by members of the editorial board and by several independent reviewers. In light of the reviews (below this email), we would like to invite the resubmission of a significantly-revised version that takes into account the reviewers' comments.

We appreciate you sending your exciting work to PLoS Computational Biology. As seen in the reviewers' reports, the work is not suitable for publication in present form. We are looking forward to receiving the revised manuscript that satisfactorily addresses reviewers comments and suggestions.

We cannot make any decision about publication until we have seen the revised manuscript and your response to the reviewers' comments. Your revised manuscript is also likely to be sent to reviewers for further evaluation.

Sincerely,

Feng Fu

Associate Editor

PLOS Computational Biology

Natalia Komarova

Deputy Editor

PLOS Computational Biology

Thank you very much for sending your exciting work to PLoS Computational Biology. As seen in the reviewers' reports, the work is not suitable for publication in present form. We are looking forward to receiving the revised manuscript that satisfactorily addresses reviewers comments and suggestions.

Reviewer's Responses to Questions

**Comments to the Authors:**

Reviewer #1: Thanks for the opportunity to review this paper. This paper proposed a new model which can nicely explain the avoidance behavior. The authors verified the validity of the simulation by systematically comparing the experimental results. This is a nice and solid study, but I still have several following comments:

Major comments:

1. The authors claim that the rule of predator-avoidance is independent of predator-prey distance. However, the analyses of the rule are actually based on the distance between predator-prey distance (e.g. Fig. 5). The closer the distance, the greater the difference in the direction from the predator. Yes, there is no direct distance as the input in the model, but the authors cannot reject the hypothesis that the direction change is related to the distance, which is clearly shown in the experimental data and simulation data. Moreover, there is a distance filter before applying this rule (e.g. dis<60), so there is at least a distance-based threshold for the rule. Overall, the distance is still a direct or indirect variable in the model.

2. It is not clear how did the author control the predator. There are several known strategies for predators, including anticipation (Mischiati et.al 2015 Nature) and proportional navigation guidance law (Brighton et.al 2017 PNAS). A direct pursue is not common in Nature. The last paper (Sankey et.al 2021 Current Biology) from these authors describes that the robot predator is controlled by a human. Commonly, human uses anticipation in the visual-based control. So, it would be reasonable to apply an anticipation-based algorithm for the predator.

3. The process of flocking after the initial turn of one agent is quite similar to the studies of phase transitions with two leaders (Couzin et.al 2005 Nature). In this paper, one “leader” is the agent who initialized the turning (predator avoidance), and the other “leader” is by the group (Group alignment). From Fig. 4, it is also clear that the phase transition is geometry (distance) based. “Group alignment” wins when the distance is larger, while “predator avoidance” wins when the distance is smaller. A compromise happens when the distance is moderate. This also reminds me of a recent paper by Sridhar et.al (2021 bioRxiv). It would be great to add a comparison of this phase transition in the experiments and simulations.

4. The analyses of Fig. 6 are similar to those complex contagions shown in fish school (Rosenthal et.al 2015 PNAS). The group behavior is not directly (fully) controlled by the initiator. They are affected by the neighbors who are affected by the initiator. Therefore, individuals are indirectly affected by the initiator. It would be nice to add a such discussion.

Minor comments:

1. Page5, line 161, the authors mentioned this \\Phi_i in Fig. 1, but there is no such symbol in Fig. 1

2. Page8, line 244, why update asynchronously?

3. Page10, Fig. 2 \\ Theta_{a_j} in the figure should be \\theta_c in the caption?

4. Why simulation in 2D, while experiments are in 3D. Add a discussion.

Reviewer #2: The current work mainly contributes to propose a data-inspired computation model that simulates and studies the self-organization collective escape in pigeon flocks. Its organization is legible and the experimental results are abundant. As a whole, the current work is convincing of the model’s reasonability and experimental results. Herein, my comments focus on the details in the proposed model,

(1) Because the proposed model is an agent-based framework, it includes a mass of parameters. The partial parameters are determined according to the empirical values in previous works and others are determined by visual testing. Does the values of these parameters (especially for these parameters determined by the visual testing) change largely with respect to the different flock sizes?

(2) Do the flock sizes dynamically evolve in the escaping process? If yes, how do the flock sizes changes with the evolving time? How does it statistically analyze the experimental results?

In addition, the non-trivial comment is that “what is the important differences of the proposed model and conclusion between the current work and the work in reference [24]”.

Reviewer #3: This manuscript developed a computational model to study the collective escape of bird flock under the attack of a single predator. The authors revealed an increase of the escape frequency when predator getting closer, which is consistent to the experimental observations. In addition, they claimed that this increased escape frequency emerges from interaction rules that is independent of predator-prey distance. The authors explained in details of the variations of the flock shape, alignment and attraction forces during the collective escape. These computational simulations are valuable to increase our understandings of the mechanism of collective escape. I would recommend the publication of this manuscript in PLOS Computational Biology after the authors have addressed the following comments:

Major comments:

• The authors performed simulations where the pigeon is modeled to sense only the predator heading, and not the predator position. These simulations showed an increased escape frequency when predator getting closer. Can the authors perform new simulations where the pigeon senses both the predator heading and predator position, or where pigeon only senses the predator position? Will these new simulations also show increased escape frequency? If so, the conclusion that pigeon senses only the predator heading may not be true.

• Although the authors showed a good agreement between the simulation results and experimental data in Figure 3, this comparison is only for one of the 43 flocks collected in the experiments. The authors also realized that there is a large variability in the measurement of real flocks. Thus, it is might be doubt whether the conclusions draw from this simulation can be applied to other pigeon flocks, e.g., flocks with larger number of birds.

• The mechanism why the escape frequency increases when the predator getting closer is not well explained. Figure 5 shows several parameters which scale with the predator-prey distance. But it is not very clear why these parameters depends on the distance. The authors explained them through reinforcement. Why reinforcement only plays a role when the predator gets closer, not for predator moves apart?

• It seems like the escape mechanism described in Figures 6 and 7 are only true for one particular case, where there are two turns in an escape and the predator changes the direction of attack when getting closer to the flock. If the predator use a different attack strategy or attack from a different direction, will the flocks still use the same escape mechanism?

Minor points:

Figure 1: The force of predator avoidance should be re-plotted such that it is perpendicular to the predator’s heading.

Line 193: The speed-attraction force in the main text is denoted as phi^cs, while it is denoted as phi^ca in Figure 1. The symbols should be consistent.

Line 309: The symbols are not consistent to Figure 2. The average heading of flock is denoted as rho_c in the main text, but rho_a in the figure. The angle between the flock heading and the individual heading is sometime denoted as theta_a, and sometime as theta_c.

Line 309: The authors distinguish between toward the flock and away from the flock by the sign of theta_a (the angle between flock heading and the individual heading). It is not clear to me how the sign of an angle is defined. What is the meaning for a negative angle?

Line 332, similarly, it is unclear how the sign of an angle is defined in Equation 7.

Line 320, (1b, 2b) should be corrected into (1b, 2a).

Figure 3 C1 & C2, why the density is highest at a distance close to 0? I expected the highest density at the nearest neighbor distance of 1.3 m.

**Have the authors made all data and (if applicable) computational code underlying the findings in their manuscript fully available?**

Reviewer #1: Yes

Reviewer #2: Yes

Reviewer #3: Yes

PLOS authors have the option to publish the peer review history of their article (what does this mean?). If published, this will include your full peer review and any attached files.

Reviewer #1: No

Reviewer #2: No

Reviewer #3: **Yes: **Hangjian Ling
---

## [Decision Letter · Decision Letter 1]

3 Dec 2021

Dear Mrs Papadopoulou,

Thank you very much for submitting your manuscript "Self-organization of collective escape in pigeon flocks" for consideration at PLOS Computational Biology. As with all papers reviewed by the journal, your manuscript was reviewed by members of the editorial board and by several independent reviewers. The reviewers appreciated the attention to an important topic. Based on the reviews, we are likely to accept this manuscript for publication, providing that you modify the manuscript according to the review recommendations.

Please kindly address the remaining comments of R#1 in your revised manuscript, so we will be able to accept it formally. Thank you.

Sincerely,

Feng Fu

Associate Editor

PLOS Computational Biology

Natalia Komarova

Deputy Editor

PLOS Computational Biology

[LINK]

Please kindly address the remaining comments of R#1 in your revised manuscript, so we will be able to accept it formally. Thank you.

Reviewer's Responses to Questions

**Comments to the Authors:**

Reviewer #1: Only two comments regarding the responses to Major comments 2 and 3:

Comment1:

The authors mentioned the robot falcon is controlled by an operator with a direct pursuit strategy. This is different from the falcon in nature (e.g., Brigthon et al. 2017 PNAS). One can naturally ask if the pigeons think the robot falcon is a predator? I think this is difficult to answer without behavior data of pigeons reacting to live falcons and robots. Without these data (even previous paper is accepted in Current Biology under the same descriptions), there might be many other hypotheses to explain these behaviors, e.g., obstacle avoidance. It would be great to add a bit of discussion in this direction.

Comment2:

Individual or collective decision-making under different options does not necessarily need leaders or those special individuals having information (see Sridhar et al., 2021 BioRixv). The system in this study exactly follows the logic described in the BioRixv paper, where the decision emerges regardless of distance sensing. In this paper, two options for the individuals are avoiding robots or flying together with other neighbors. How the individuals choose one of the options is the main question answered in this study. This question is exactly the same question raised in the BioRixv paper.

Reviewer #2: The reply addresses the provided comments, and is revised in the new edition of manuscript. Thus, I can recommend it for acceptance.

Reviewer #3: The authors have fully addressed my comments. I agree the publication of this manuscript.

**Have the authors made all data and (if applicable) computational code underlying the findings in their manuscript fully available?**

Reviewer #1: Yes

Reviewer #2: Yes

Reviewer #3: Yes

PLOS authors have the option to publish the peer review history of their article (what does this mean?). If published, this will include your full peer review and any attached files.

Reviewer #1: **Yes: **Liang Li

Reviewer #2: No

Reviewer #3: No

Figure Files:

Data Requirements:

Reproducibility:

References:

---

## [Editor Report · Decision Letter 2]

19 Dec 2021

Dear Mrs Papadopoulou,

We are pleased to inform you that your manuscript 'Self-organization of collective escape in pigeon flocks' has been provisionally accepted for publication in PLOS Computational Biology.

Best regards,

Feng Fu

Associate Editor

PLOS Computational Biology

Natalia Komarova

Deputy Editor

PLOS Computational Biology

---

## [Editor Report · Acceptance letter]

5 Jan 2022

PCOMPBIOL-D-21-01142R2 

Self-organization of collective escape in pigeon flocks

Dear Dr Papadopoulou,

I am pleased to inform you that your manuscript has been formally accepted for publication in PLOS Computational Biology. Your manuscript is now with our production department and you will be notified of the publication date in due course.

With kind regards,

Agnes Pap
